# Importance of Modulating Kynurenic Acid Metabolism—Approaches for the Treatment of Dementia

**DOI:** 10.3390/biom15010074

**Published:** 2025-01-06

**Authors:** Halina Baran, Marcelin Jan Pietryja, Berthold Kepplinger

**Affiliations:** 1Karl Landsteiner Research Institute for Neurochemistry, Neuropharmacology, Neurorehabilitation and Pain Therapy, 3362 Mauer-Amstetten, Austria; berthold.kepplinger@neuro-lab.eu; 2Neurophysiology Unit, Department of Biomedical Sciences, University of Veterinary Medicine Vienna, 1210 Vienna, Austria; 3St. Francis Herbarium, Monastery of the Franciscan Friars Minor, 40-760 Katowice, Poland; jpprince@op.pl; 4Department of Neurology, Neuropsychiatric Hospital, 3362 Mauer-Amstetten, Austria

**Keywords:** kynurenic acid, xanthurenic acid, dementia, plaque, cerebrolysin, D-cycloserine, glial depressing factor, Jerusalem Balsam, herbs, *Helix pomatia* snail, memory model, anti-dementia drug, bird droppings

## Abstract

In this article, we focus on kynurenic acid metabolism in neuropsychiatric disorders and the biochemical processes involved in memory and cognitive impairment, followed by different approaches in the fight against dementia. Kynurenic acid—a biochemical part of L-tryptophan catabolism—is synthesized from L-kynurenine by kynurenine aminotransferases. Experimental pharmacological studies have shown that elevated levels of kynurenic acid in the brain are associated with impaired learning and that lowering kynurenic acid levels can improve these symptoms. The discovery of new compounds with the ability to block kynurenine aminotransferases opens new therapeutic avenues for the treatment of memory impairment and dementia. The newly developed *Helix pomatia* snail model of memory can be used for the assessment of novel pharmacological approaches. Dietary supplementation with natural molecular/herbal extracts, exercise, and physical activity have significant impacts on endogenous pharmacology by reducing kynurenic acid synthesis, and these factors are likely to significantly modulate steady-state biological conditions and delay the negative consequences of aging, including the onset of pathological processes.

## 1. Tryptophan Metabolism via the Kynurenine Pathway

### 1.1. Kynurenine Metabolism

In tryptophan catabolism, one of the pathways of particular biological interest leads to the formation of niacin, namely the kynurenine pathway [1,2] (Figure 1). The kynurenine pathway is also the main pathway for tryptophan catabolism, as approximately 90% of tryptophan is metabolized along this pathway, and only a small amount of tryptophan is converted to serotonin and melatonin. The first step in the catabolism of tryptophan by this pathway is cleavage of the indole ring. This reaction is catalyzed by two different enzymes—tryptophan 2,3-dioxygenase or indolamine 2,3-dioxygenase—resulting in the formation of N-formyl-kynurenine, which is further metabolized by formamidase to L-kynurenine.

Catabolism of L-kynurenine along this pathway has a remarkable function, namely the synthesis of the neuroactive metabolites 3-hydroxykynurenine [3], quinolinic acid [4], and kynurenic acid (KYNA) with neurotoxic, excitatory, and inhibitory properties, respectively [4,5]. L-kynurenine crosses the blood–brain barrier well, is actively accumulated in various tissues (e.g., brain, heart, and liver), and is taken up by various cells (e.g., astrocytes, neurons, and macrophages), including their organelles (mitochondria) [6,7,8]. In the kynurenine pathway, L-kynurenine is degraded to quinolinic acid by kynurenine 3-hydroxylase, followed by kynureninase and 3-hydroxyanthranilic acid oxygenase [5]. Quinolinic acid is then metabolized by quinolinic acid phosphoribosyltransferase to nicotinic acid mononucleotide and, finally, to the active coenzyme nicotinamide adenine dinucleotide (NAD) [5]. The other possibility of L-kynurenine degradation is the formation of KYNA by irreversible transamination of L-kynurenine [8,9,10] or the formation of anthranilic acid by kynureninase [5], followed by hydroxylation of anthranilic acid to 3-hydroxyanthranilic acid [11]. Importantly, the transaminase also catalyzes the synthesis of xanthurenic acid from 3-OH-kynurenine [12,13]; see Figure 1.

KYNA synthesis from tryptophan using indole-3-pyruvic acid has also been described [14,15]; however, its significance remains to be elucidated.

### 1.2. Kynurenine Aminotransferases

The capacity for the formation of KYNA from L-kynurenine by kynurenine amino-transferase (KAT) is very high, and several KATs have been discovered in different mammalian organs. In rat and human peripheral tissues, four types of proteins are capable of catalyzing the kynurenine-2-oxoacid transamination reaction to form KYNA [16,17]. This protein, called kynurenine aminotransferase (KAT), is a pyridoxal 5′-phosphate-dependent enzyme and occurs in four KAT isoforms, arbitrarily named KAT I, KAT II, KAT III, and KAT IV [18]. High activity of these enzymes has been observed in rat and human livers [9]. However, very low activity has been reported in peripheral tissues from piglets [19] or the snail *Helix pomatia* [20], suggesting species-related differences. The central nervous system (CNS) is significantly involved in KYNA formation, and three proteins have been identified in human brain tissue: KAT I [21], KAT II [22], and KAT III [23,24].

The knowledge of KATs is vast and complex, and significant progress has been made in terms of their genetic characterization, biochemical and physiological function, and structural properties [18,23,25]. Differences in substrate specificity, pH profiles, and inhibition between the four KATs have been demonstrated; for example, with respect to pH, KAT I has its highest activity at pH 9.6, which may be particularly important in pathological conditions [22,25,26], whereas KAT II—which has a neutral optimal pH of 7.4—may operate essentially under physiological conditions [22]. Furthermore, KAT III, which has an optimal pH range of about 8.0–9.0 [23,25], may share its action between physiological and pathological conditions. There are data suggesting that human KAT I is a protein with multi-functional activities and may also be an important protein for KYNA synthesis under physiological conditions. Han et al. made an important observation by studying the pH profile of the enzyme KAT I [25]. They showed that KAT I has a broad optimal pH range and is able to efficiently catalyze kynurenine to KYNA under physiological conditions, in contrast to the published pH profile of KAT I [22,24]. The inability of KAT I to catalyze efficient transamination reactions in previous studies was probably due to inhibition of the enzyme by Tris buffer in the reaction mixture, as pointed out by the authors. Therefore, it is reasonable to propose—as also pointed out by the authors—that KAT I may be an important player in KYNA synthesis under physiological conditions in the human brain. However, further research is needed to fully understand its overall physiological role in vivo.

Interestingly, KATs appear to have the ability to change their chemical properties (and, presumably, their actions) under physiological and pathological conditions. Crystallographic studies of KAT I and KAT II from different organisms, including humans, have revealed different structural features of KAT I and KAT II [27]. The authors suggested that different conformational changes during catalysis create different active sites in the two isozymes and affect their substrate specificity and that these changes may influence the ability to form KYNA. The authors also suggested that structural studies reveal interesting evolutionary relationships that may pave the way for new molecules enabling the modulation of KYNA synthesis. KYNA formation has been shown to occur preferentially in glia, astrocytes, and, to a lesser extent, neurons [28,29].

## 2. Kynurenic Acid Metabolism Across the Lifespan

KYNA metabolism shows a characteristic pattern of changes throughout the lifespan of various animal species, including humans. In the rat and sheep CNS, KYNA increased during the embryonic stage until the day of birth, then decreased dramatically [30,31]. In rats, there was also a low level of KAT activity during the first week after birth [32], which slowly and progressively increased during ontogeny, maturation, and aging [32,33]. The dramatic decrease in KYNA on the day of birth [30,31] suggests that KYNA is involved in the physiological processes associated with the birth of a child.

Consistent with an increase in KYNA throughout life [34], an increase in L-kynurenine—the bioprecursor for KYNA synthesis—has been reported in the cerebrospinal fluid of aged rats [35] as well as in the cerebrospinal fluid of aged humans [36]. Tryptophan has also been found to increase with age [36]. The high levels of tryptophan in the elderly may be due to uncontrolled consumption of foods such as chocolate but also to a reduced capacity for biochemical degradation during aging. The remarkable change in KYNA metabolism during ontogeny and maturation in the mammalian brain has been suggested to be a consequence of the development of neuronal connectivity organization, synaptic plasticity, and receptor recognition [5]. It is questionable whether KYNA has different functions throughout life, considering the currently available information on the neuronal network.

Tryptophan metabolism also plays a central role in the gut–brain axis [37]. Kronsteiner (one of my collaborators), in her Master’s thesis on the biology of tryptophan metabolites in the golden eagle (Aquila chrysaetos), showed the significant presence of tryptophan, kynurenine, KYNA, xanthurenic acid, anthranilic acid, 3-hydroxykynurenine, 3-hydroxyanthranilic acid, and quinolinic acid in fecal samples from the birds [38,39]. Furthermore, a study of the aging process showed that KYNA, xanthurenic acid, and quinolinic acid levels in feces decreased with increasing age of the eagle between 1 and 18 years [38]. KYNA was not affected by sex or season. Other tryptophan metabolites increased with age. We believe that the reduced levels of KYNA and xanthurenic acid in bird feces with age may reflect the importance of the digestive tract for good memory in birds as well as maintaining a healthy brain; this is likely also true in humans in general. 

## 3. Kynurenic Acid and Neuroprotection

### 3.1. Action of Kynurenic Acid

In the final decades of last century, important information about the action of KYNA was revealed. KYNA not only acts as an endogenous antagonist of the glutamate ionotropic excitatory amino acid receptors N-methyl-D-aspartate (NMDAR), alpha-amino-3-hydroxy-5-methylisoxazole-4-propionic acid, and kainate [5,40] and the nicotinic acetylcholinergic subtype alpha-7 receptor (α-7nAChR) [41] but is also an agonist of the orphan G protein-coupled receptor GPR-35 [42,43] and has anticonvulsant and neuroprotective activities [44,45]. Regarding the importance of the involvement of KYNA and xanthurenic acid in relation to the receptor GPR-35 in the gut microbiota, specific GPR-35-positive signals have been detected in the gastrointestinal tract [42], which should be further explored.

In the brain, KYNA levels range from low nanomolar to low micromolar concentrations [5,46], and there is evidence that these physiologically relevant concentrations of KYNA are likely to block α-7nACh more effectively than NMDA receptors [47,48]. With respect to KYNA inhibition of glutamatergic neurotransmission, accumulated data suggest that the glycine co-agonist site of the NMDA receptor is not saturated [5,49,50], so physiological concentrations of KYNA may be sufficient to inhibit NMDA receptor activity.

With regard to the inhibition of cholinergic neurotransmission, KYNA inhibits α-7nAChRs non-competitively (IC 50 approximately 7 µM) [41,47,48]; the inhibition of α-7nAChRs by KYNA is likely to be voltage-independent [47,48]. Speculatively, it has been suggested that KYNA may act via intracellular second messengers that affect the function of α-7nAChR [48]. Interestingly, KYNA could also increase the expression of non-α-7nAChRs, α-4-ß-2nAChRs [49], suggesting a notable interaction between KYNA and the receptor functions of cholinergic neurons. Importantly, α-7nAChRs play multiple roles in modulating the glutamatergic system in both normal and diseased nervous systems [50,51]. On the other hand, a biphasic change in KYNA formation has been reported after prolonged nicotine application [52], suggesting its down- and/or up-regulation by nicotine.

Regarding the mechanism of action of KYNA on cholinergic neurons, the work published by Stone [53] showed that there is no confirmed, reliable evidence of antagonistic activity by KYNA at the nicotinic receptor; therefore, the results should be interpreted only in terms of the confirmed site of action, that is, the blocking effect on glutamatergic neurons. These comments by Stone [53] also suggest that the experimental conditions of the studies are unlikely to be exactly the same.

### 3.2. Neuroprotection of Kynurenic Acid

The neuroprotective and anticonvulsant activities of KYNA have been documented [44,45], with this role most likely related to the modulation of both glutamatergic and acetylcholinergic neurotransmission, as the overall effects of NMDA receptor antagonists and the effects of α-7nAChR antagonists on neuronal plasticity and viability are similar and close to those of KYNA, as demonstrated via different pharmacological approaches. For example, KYNA and α-7nAChR antagonists can block neurite outgrowth [54,55], reduce apoptotic neuronal death [54,56], and have anticonvulsant activity [57,58]. Accumulating data suggest that both NMDA and nACh receptors are involved in the regulation of neuronal activity and survival in the brain.

An interesting and complex study of KYNA neuroprotection has been presented by Klein et al. [59], in which the authors showed that low micromolar concentrations of KYNA strongly increased neprilysin gene expression, protein levels, and enzymatic activity in human neuroblastoma SH-SY5Y cells. Neprilysin is a well-characterized metallopeptidase whose deregulation leads to cerebral Aβ accumulation and neuronal death in Alzheimer’s disease [59]. Furthermore, the authors demonstrated that KYNA exerted a protective effect on SH-SY5Y cells by increasing their viability through an NMDA receptor-independent mechanism. However, KYNA-induced neuroprotection was abolished in the presence of thiorphan, an inhibitor of neprilysin activity. The data presented demonstrate the complexity of the mechanism uncovered under experimental conditions. We suggest that lowering the KYNA concentration might play a pivotal function, affecting the probability of the induction of pathological events in some places.

The antioxidant activity of KYNA has been suggested to play a role in neuroprotection, as shown by Bratek-Gerej et al. [45]. KYNA-mediated neuroprotection was observed in a neonatal hypoxia–ischemia model, and the authors suggested that it was largely related to the reduction of oxidative stress.

### 3.3. Anticonvulsant Drugs and Kynurenic Acid Synthesis

An interesting observation has been made when studying the effect of anticonvulsant drugs on KYNA metabolism in rat cortical slices. Kocki et al. observed an increase in KYNA formation in the presence of these drugs in vitro [60]. The activities of the biosynthetic enzymes KAT I and KAT II also increased. Importantly, not all anticonvulsants had a similar effect. Drugs such as phenobarbital, felbamate, phenytoin, and lamotrigine enhanced KYNA formation in vitro and stimulated the activity of KAT I, whereas vigabatrin, gabapentin, and tiagabine inhibited KYNA synthesis in cortical slices and also reduced the activities of KAT I and KAT II. The authors noted that the enhancement of KYNA by some antiepileptic drugs suggests a novel mechanism, which acts in a synergistic manner with other actions of these drugs. This finding is potentially valuable in terms of better control of epilepsy and neuroprotection. However, it is reasonable to ask whether the memory impairment observed in some epileptic patients could be due to chronic use of such anticonvulsant drugs, which increase KYNA formation.

Despite remarkable progress in understanding neuropsychiatric and neurodegenerative disorders, there has been limited success in developing neuroprotective agents that can slow disease progression and prevent neuronal death or act as anti-dementia agents. Ostapiuk et al. [61] indicated that the current state of clinical research is to develop promising experimental methods to manipulate the kynurenine pathway in humans without side effects, with the goal of inhibiting the part of the kynurenine pathway responsible for quinolinic acid synthesis while stimulating KYNA synthesis. Potential treatments may include inhibitors of specific enzymes in the kynurenine pathway as well as new drugs and analogues of KYNA that can cross the blood–brain barrier and enhance KYNA-induced blockade of the glycine site within the NMDA receptor complex [61,62].

## 4. Dysfunction of Kynurenic Acid Synthesis and Its Significance

### 4.1. Decrease or Increase in Kynurenic Acid

A dysfunction in KYNA synthesis (i.e., a decrease and/or increase in KYNA production in the brain) has been identified as an important event contributing to neurodegeneration. In vivo, experimental studies have shown that reducing KYNA synthesis in the adult rat brain using non-specific inhibitors led to neurotoxic effects [63,64]. Interestingly, a preferential loss of layer III of the entorhinal cortex after local injection of a non-specific inhibitor of KYNA synthesis has been shown to be an important factor in the pathophysiology of human temporal lobe epilepsy [65,66]. On the other hand, increases in brain and serum KYNA levels have been observed in rats exposed to kainic acid-induced stereotypic behavior and seizures [67] during oxygen deprivation [68] or after encephalomyocarditis virus (EMCV) infection in piglets [69].

### 4.2. Kynurenic Acid and Lethality

Accumulated data suggest that KYNA is a very sensitive biochemical metabolite of the kynurenine pathway, which has been observed in various experimental pathological events. For example, we observed a dramatic increase in brain KYNA levels during a 15–20 min period of asphyxia, characterized by almost 100% lethality [68]. Herrera-Marschitz et al. have reported that a lack of oxygen lowers nicotinamide adenine dinucleotide tissue stores and decreases ATP formation in the brain and heart, weakening the electron transport pump, anaerobic metabolism, and acidosis, necessarily leading to death if oxygenation is not re-established [70]; these chemical events may be relevant in stroke [71,72] or even sudden death.

Other experimental work in vivo has shown that administration of KYNA (icv) to rats resulted in ataxia and stereotypy in a dose-dependent manner (0.025–1.6 µmol) [73,74]. Importantly, the authors demonstrated that administration of 0.8 µmol KYNA resulted in drowsiness and approximately 25% animal mortality, and, at a dose of 1.6 µmol, all animals died of cardiorespiratory failure within 2–5 min [73,74]. These data suggest that marked synthesis of KYNA in the brain and periphery due to infection and/or other causes might affect more organs, leading to various symptoms, memory impairment, and cardiorespiratory dysfunction, followed by acute death. Along this line, data from EMCV-infected piglets demonstrated activation of the kynurenine metabolic pathway in the periphery and in the brain, accompanied by lethality [69].

### 4.3. Tryptophan Metabolites Affect Mitochondrial Respiratory Parameters

Interestingly, differences in the effects of tryptophan metabolites on mitochondrial respiratory parameters in the heart, brain, and liver have been observed. We have reported significant dose-dependent effects of KYNA, 3-OH-kynurenine, xanthurenic acid, or 3-OH-anthranilic acid on respiratory parameters in rat brain, heart, and liver mitochondria [75]. Furthermore, high levels of KYNA increased mitochondrial oxygen consumption and decreased ATP synthesis in heart mitochondria but not in brain or liver mitochondria, whereas 3-OH-kynurenine, 3-OH-anthranilic acid, and xanthurenic acid affected mitochondrial respiratory parameters in the heart, liver, and brain. Although the present study does not support the assumption of a correlation between impairment of brain mitochondria and high KYNA levels, it is important to mention that enhanced kynurenine aminotransferase activities in the brain may also increase the formation of xanthurenic acid from 3-OH-kynurenine. Importantly, in the presence of xanthurenic acid, the respiratory control (RC) value of brain mitochondria was significantly lowered, as oxygen consumption during the passive state was slightly increased, whereas oxygen consumption during the active state was moderately lowered in the presence of glutamate/malate. These data suggest the impairment of brain mitochondria in the presence of increased xanthurenic acid levels. No effect of mM quinolinic acid doses on mitochondrial respiratory parameters in rat heart, liver, or brain was detected. The aging process did not affect the quality of mitochondrial respiratory parameters in the brain, heart, and liver, at least in healthy rats.

These findings suggest that activation of L-kynurenine metabolism may affect various organs, particularly brain and cardiac cell function, leading to impairment of memory and cardiovascular processes as well as congestive cardiac dysfunction.

## 5. Kynurenic Acid, Neuropsychiatric Disorders, and Dementia

### 5.1. Loss of Cholinergic Neurons

In the early 1970s, Olney, Rothman, Choi, and others showed that excessive activation of NMDA receptors—a cellular phenomenon involved in neuronal signaling in both neurotrophic and neuronal plasticity—can trigger a series of intracellular and/or extracellular events associated with neuronal death [5]. At the same time, this phenomenon was thought to provide a mechanism for memory formation in the brain. According to the hypothesis proposed by Greenmayer et al. in 1988 [76] and supported by others, glutamatergic transmission plays an important role in the neuropathological mechanism and symptomatology of dementia [5,66].

Whitehouse et al. (1981) showed that the nucleus basalis of Meynert provides a diffuse cholinergic input to the neocortex [77]. Compared to an age- and sex-matched control, the nucleus basalis of an AD patient showed a significant loss of neurons. The loss of this neuronal population was an anatomical correlate of the well-documented cholinergic dysfunction in AD [77]. Neurochemically, a reduction in choline acetyltransferase has emerged as an important marker of dementia [78]. Loss or impairment of cholinergic neurons has been described in Alzheimer’s disease [79,80], vascular dementia [81], and Down syndrome [82] and in association with aging [78,81]. In particular, cognitive impairment in adults with Down syndrome resembles early cognitive changes in AD [83].

### 5.2. High Kynurenic Acid Level and Learning Impairment

Experimental pharmacological studies have shown that KYNA and 5,7-dichlorokynurenic acid improved social and object recognition in rats [84], whereas 7-chlorokynurenic acid impaired memory formation [85] and increased spatial memory deficits induced by endogenous KYNA levels [86]. These data suggest that KYNA metabolism plays a role in human memory and cognitive impairment.

Elevated brain levels of KYNA have been reported in patients with early-onset Huntington’s disease [87], Parkinson’s disease [88], Down syndrome [89], Alzheimer’s disease [90], and schizophrenia [91] as well as in HIV-1-infected patients [92]. Elevated KYNA levels in CSF have also been found in HIV-1-infected patients [93], schizophrenia patients [94], hydrocephalus patients [36], children with cerebral malaria [95], patients with the acute phase of multiple sclerosis [96], and during aging [34].

Reduced CSF KYNA levels have been reported in advanced multiple sclerosis by both our group and others [96,97,98]. Along this line, we observed reduced KYNA metabolism and lowered activity of neuronal markers of GABA and cholinergic neurons in the brain via post-mortem material [99]. These data suggest that the balance between inhibitory and excitatory activities in the brain is significantly damaged in advanced multiple sclerosis patients.

Interestingly, a study comparing dementia in patients with multiple sclerosis and Alzheimer’s disease revealed differences in the analysis of deviation scores, controlling for the overall level of cognitive impairment [100]. The authors described that both groups showed significant impairment on the test battery, but the degree of dementia was more severe in the patients with Alzheimer’s disease. Alzheimer’s disease was associated with relatively greater impairment of learning, memory, and verbal skills, whereas the MS group showed greater relative impairment of attention, incidental memory, and psychomotor functions.

The presented data support a distinction between “gray matter” and “white matter” dementia, which could help to clarify the issue of “cortical” vs. “subcortical” dementia, with differences in biochemical markers possibly supporting the distinction [99].

Cognitive and memory impairments and elevated KYNA levels are common to most neuropsychiatric and infectious diseases, and our research has focused on exploring whether modulating KYNA synthesis can be considered as part of the way to ameliorate these symptoms.

### 5.3. Alzheimer’s Disease

Increased KYNA levels in Alzheimer’s disease, ranging from 123 to 192% of CO [90], have been observed in the frontal cortex, hippocampus, putamen, caudate nucleus, and cerebellum. A study focused on KAT I and KAT II revealed a significant increase in KAT I in the putamen and caudate nucleus, while KAT II was moderately affected; see Figure 2, adapted from [90]. KAT I in the basal ganglia regions plays a key role in KYNA elevation under pathological conditions. Could the dopamine deficit in the putamen and caudate nucleus [101] be the critical player and a cause for the enhancement of KATs or KYNA synthesis? The presented data suggest that the increases in KYNA in the frontal cortex, hippocampus, and cerebellum are not due to changes in enzyme activity but instead due to other causes, such as less-effective clearance of KYNA.

Several other abnormalities have been reported in the activation of the kynurenine pathway of tryptophan catabolism in Alzheimer’s disease, such as the activation of indolamine 2,3-dioxygenase, the first and rate-limiting step of the pathway. Zádori et al. have suggested that the metabolites 3-hydroxykynurenine, 3-hydroxyanthranilic acid, and quinolinic acid are involved in promoting glutamate excitotoxicity, ROS production, lipid peroxidation, and microglial neuroinflammation [102].

The involvement of quinolinic acid in the pathogenesis of Alzheimer’s diseases—particularly regarding the enhancement of its synthesis by human macrophages and microglia—has been intensively studied by Guillemin et al. [103]. The authors also paid attention to the modulatory effects of quinolinic acid and KYNA on the amyloid peptide 11–42 and 1–40 aggregation and suggested quinolinic acid as a significant factor involved in the pathogenesis of neuronal damage in Alzheimer’s disease [104].

### 5.4. Parkinson’s Disease

Significantly elevated KYNA levels have also been reported in PD patients with dementia but not in those without dementia [88]; these observations support the idea that KYNA is important in the context of dementia. Decreased dopaminergic neurotransmission of the nigrostriatal loop is characteristic of the aging process [101], and the progression of this reduction is characteristic of PD [105]. The interaction between dopamine reduction in the striatum and high levels of KYNA in the basal ganglia [106] is a curious mechanism of the aging process, which may break down under certain circumstances, causing the changes that are characteristic of Parkinson’s disease.

Some interesting findings and correlations between dopamine and KYNA can also be drawn from experimental data. The reduction of dopamine levels in various rat brain regions, such as the limbic and basal ganglia regions, in the kainic acid epileptic rat model [107] was accompanied by a significant increase in KYNA levels [67]. Kainic acid-treated rats also developed a memory deficit. Interestingly, the levels of the neurotransmitter serotonin did not change during the acute phase of kainic acid-induced epilepsy, although there was a marked increase in the turnover of both catecholamine and serotonergic neurons.

### 5.5. Down Syndrome

The study of KYNA metabolism in Down syndrome requires some clarification. A pronounced increase in KYNA levels (Figure 3, adapted from [89]) but decreased KAT I activity and normal KAT II (Figure 4, adapted from [89]) has been measured in the frontal and temporal cortex of Down syndrome subjects. These data indicate a remarkable discrepancy between high KYNA levels and low KAT activities in both cortices [89], which is difficult to explain. At the time of the Down syndrome study, there was no knowledge of increased KYNA synthesis in the brain of patients with bronchopneumonia. According to the clinical findings, the control patients, but especially the Down syndrome patients, had pathologies (i.e., infarction or bronchopneumonia) that are characterized by significantly increased KYNA levels. This could be the reason for the high KYNA levels observed in the frontal and temporal cortex in Down syndrome patients with bronchopneumonia [89].

Importantly, it is questionable whether increased KYNA metabolism in the brain of patients with Down syndrome may also contribute to their reduced life expectancy in general. The average life expectancy of patients with Down syndrome is around 40–50 years, and these patients have high levels of KYNA (over 300% vs. CO) in the brain. Bronchopneumonia in Down syndrome can lead to oxygen desaturation and high levels of KYNA in the brain, resulting in early death. Recently, a higher incidence of sudden death due to coronavirus infection has been reported in patients with Down syndrome [108]. Therefore, the involvement of high KYNA levels in the brain tissue of patients with Down syndrome may be relevant to the incidence of high lethality.

Considering the above, it is recommended that a new study of KYNA metabolism in patients with Down syndrome using good human post-mortem material should be undertaken.

### 5.6. HIV-1 Virus Infection

Neuropsychiatric symptoms, often observed in patients infected with the HIV-1 virus, are also associated with a marked increase in brain KYNA metabolism [92]. A marked increase in brain KYNA levels after HIV-1 infection (Figure 5, adapted from [92]) correlated significantly with high KAT I activities, whereas KAT II was only moderately altered (Figure 6, adapted from [92]). The data indicate that KAT I is the most affected enzyme in HIV-1 infection, supporting the idea that KAT I is predominantly involved in KYNA synthesis under pathological conditions.

Increased KYNA metabolism and neuropsychiatric symptoms have also been reported in piglets following EMCV infection [69].

### 5.7. Microglia Activation

Microglial activation is an important pathological feature of a variety of neurological disorders, including aging, and is correlated with increased KYNA metabolism. There is a predominant astrocytic localization of KAT in the rat brain [16,28], and the major intermediate filament component of astrocytes—glial fibrillary acidic protein (GFAP)—increases with age [109,110], similar to KAT activity [16]. The presence of glia-depressing factor (GDF) has been suggested [111], which has KAT-blocking properties and is thought to exert the opposite effect of GFAP. We observed a significant impact of GDF on KYNA metabolism in multiple sclerosis patients, likely affecting glial/astroglial activity and glial proliferation.

The interactions between glial and immune cells and the secretion of various cytokines, such as interferons, interleukins-1 and -6, and tumor necrosis factor, in response to injury or infection or during the aging process play a prominent role in the initiation and propagation of CNS damage [112,113]. Activation of tryptophan degradation by tryptophan 2,3-dioxygenase via interferon has been observed in human monocytes/macrophages as well as a variety of human cells and cell lines in vitro [114,115]. The positive correlation between increased CSF β2-microglobulin (a marker of immune reactivity) and KYNA levels during the aging process suggests the activation of immune cells [34]. Increased CSF β2-microglobulin levels have been reported in Alzheimer’s disease [116] and HIV-1 infection [93,117]. However, decreases in ß-microglobulin are likely to be associated with the development of events leading to mortality. An association between lower serum β2-microglobulin levels and worse survival has been described in peritoneal dialysis patients [118]. There was also a good correlation between low β2-microglobulin and low mortality in piglets infected with EMCV on day 4 [69].

## 6. Different Types of Pathology After HIV-1 Infection

### 6.1. Pathology

There are substantial data demonstrating that several infections, including HIV-1, are associated with the development of dementia. The dominant clinical features include deficits in cognitive processing speed, concentration, attention, and memory. In 1993, Budka noted that previous data on the incidence and severity of HIV-associated dementia appeared to be somewhat biased by experience with a selected clinical population and the lack of generally accepted criteria for such a diagnosis; in particular, the term “dementia” appeared to be used more broadly in the United States than in Europe [119]. Recent research on the assessment of CNS status in HIV patients, including neuropsychological testing and neuroimaging, has focused on improving the reliability of ruling out comorbid conditions and allowing for earlier diagnosis of dementia [120]. In addition, HIV patients should be regularly screened for cognitive impairment [121].

Morphologic studies of the cerebral cortex—including correlations with clinical and neuropathologic data—have clearly shown that AIDS patients with dementia have a significantly greater loss of neurons in the frontal cortex when compared to those without dementia [119]. In addition, immunohistochemical and morphometric studies have shown increased numbers of GFAP-positive astroglia in the cerebral cortex of HIV-positive patients with and without clinical and/or pathological evidence of encephalopathy [119,122]. These data correlate well with the observed increase in KAT activity in the frontal cortex of HIV-1-infected patients [92].

### 6.2. Kynurenic Metabolism and Different Pathologies

In a retrospective study, we evaluated KYNA metabolism in the brain under different pathologies in order to better understand the biochemical changes associated with different pathologies [123]. The types of pathology in HIV-1 patients were classified as HIV in the brain (HIV), opportunistic infection (OPP), cerebral infarction (INF), malignant lymphoma of the brain (LY), and glial dystrophy (GD) [119].

Of the pathologies studied, OPP was the most common (65%), followed by HIV (26%), then LY, INF, and GD (22% each). In addition, 68% of HIV-1 patients had bronchopneumonia, with the highest incidence of 60% in the OPP and LY groups [123].

KYNA was significantly increased in the frontal cortex in LY (392% of CO), HIV (231% of CO), and GD (193% of CO) as well as in the cerebellum in GD (261% of CO). The elevation of KYNA strongly suggests its involvement in dementia associated with all pathologies.

L-kynurenine—a bio-precursor of KYNA—was also elevated in the frontal cortex of LY (385% of CO) and INF (206% of CO) as well as in the cerebellum of GD, LY, OPP, and HIV (between 177 and 147% of CO). In good correlation with high KYNA formation, the KAT enzyme levels were also enhanced. KAT I activity was significantly increased in the frontal cortex of all pathological subgroups (OPP = 420% > INF > LY > HIV > GD = 192% of CO). Similar changes were found in the cerebellum, where KAT I activity was significantly increased in all pathological subgroups (OPP = 320% > LY, HIV > GD > INF = 176% of CO). KAT II activity, in contrast, was only moderately but significantly higher in the frontal cortex of INF and OPP. Furthermore, in the cerebellum of HIV, OPP, and LY, it was comparable to that of controls, while in INF and GD, it was even slightly reduced. These pathological conditions are in line with the glial proliferation induced by quinolinic acid-induced neurodegeneration [5,119,124]. Correlation analyses between kynurenine parameters revealed an association between a high KAT I/KAT II ratio and increased KYNA levels and lower L-kynurenine in the frontal cortex and cerebellum of the HIV and LY subgroups.

We found that OPP was the most common pathological subgroup, with the highest incidence after infection, followed by significantly activated KYNA metabolism and a predominant increase in KAT I. Bronchopneumonia was common, at 70%, in HIV-1-infected patients. Furthermore, among the normal subjects, we found some who had been diagnosed with bronchopneumonia, who were characterized by significantly high KYNA metabolism in the brain. KAT I activity in the frontal cortex and cerebellum increased by approximately 877% and 479% of CO, respectively [123]. This finding suggests a remarkable correlation between impaired conditions of oxygen availability and increased KYNA formation in the human brain. These observations may also have implications for understanding the pathological processes in the brain following HIV-1 infection as well as other infections associated with the development of neuropsychiatric and neurological symptoms, including memory and cognitive impairment as well as lethality.

Our data on KYNA are in line with the marked increase in KYNA in cerebrospinal fluid reported by Heyes et al., who presented the important finding that quinolinic acid—an endogenous neurotoxin that is a metabolite of the kynurenine pathway—was markedly elevated to toxic concentrations in the cerebrospinal fluid of HIV-infected patients [93,124]. This increase in quinolinic acid is the main cause for the development of neurodegeneration and complex pathology.

In humans, respiratory distress, bronchopneumonia, lobar pneumonia, pulmonary edema, or even tuberculosis can occur after various infections such as those by HIV-1 [124,125,126], coronavirus [127], influenza A [128], or EMCV [69,129]. High mortality has also been reported in infected mammals, including humans [130]. Along this line, the study conducted by Murray on the occurrence of pathological conditions and related causes of death showed cardiovascular diseases, respiratory infections, HIV/AIDS, tuberculosis, malaria, asphyxia, and ischemia as significantly frequent causes of high mortality [131]. We believe that high levels of KYNA synthesis, along with the presence of cognitive and memory impairment, are an additional significant factor affecting mortality. Anti-dementia treatments in HIV patients have, unfortunately, not been successful. To date, several non-antiretroviral therapies, including minocycline, memantine, selegiline, lithium, valproic acid, nimodipine, rivastigmine, and others, have been investigated for the treatment of HIV-associated dementia, without significant results [132]. With these data, it is reasonable to ask whether lowering KYNA levels could contribute to the improvement of these symptoms, as classical anti-dementia drugs have not been successful in this task. It is possible that the daily use of a herbal drink can minimize the symptoms and development of the disease after HIV-1 infection, as has been observed recently in the context of COVID infection [133,134]. Therefore, measuring KYNA levels in the cerebrospinal fluid and/or serum of HIV patients would be particularly valuable, not only clinically but also scientifically.

## 7. Anti-Dementia Approaches

### 7.1. Classical Anti-Dementia Drugs

Several approaches have been approved to treat dementia or improve memory and cognition, including drugs that interact with cholinergic activities (i.e., acetylcholinesterase inhibitors, such as galantamine, donepezil, and rivastigmine), drugs that block glutamate neurotransmission (e.g., NMDA receptor blockers, such as memantine), nootropics (e.g., piracetam), ginkgo biloba, and cerebrolysin, among others. Tan et al. provided evidence that cholinesterase inhibitors and memantine were able to stabilize or slow the decline in cognition and function [135], with cholinesterase inhibitors showing a modest overall benefit for slowing behavioral decline and clinical global change in patients with AD. However, compared with placebo, there were more discontinuations and adverse events with the cholinesterase inhibitors but not with memantine, which may reflect the good safety profile and greater tolerability of memantine. In another study, Hansen et al. showed that the cholinesterase inhibitors donepezil, galantamine, and rivastigmine were also able to halt or slow declines in cognition, function, behavior, and global change compared with placebo [136]. There is no clear evidence to determine whether one of these drugs is more effective than another, although adjusted indirect comparisons suggest that donepezil and rivastigmine may be slightly more effective than galantamine, at least as reflected in some outcome measures.

A significant interaction between KYNA action and/or metabolism and galantamine treatment has been demonstrated; the mechanism of action of galantamine in the treatment of dementia has been suggested to be due to its action as an allosteric potentiating ligand of nicotine as well as a competitive antagonist of KYNA-induced inhibition of α-7nACh receptors [137]. The development of this relationship has been described in the review by Stone et al. (2014) [43]. Piracetam, a widely used nootropic drug, has been hypothesized to improve memory function through its influence on synaptic plasticity and neurotransmitter levels. In 2001, Flicker pointed out that, in order to determine the clinical efficacy of piracetam for the features of dementia or cognitive impairment, it should be classified according to the major subtypes of dementia, namely vascular, Alzheimer’s disease, or mixed vascular and Alzheimer’s disease, or unclassified dementia, or cognitive impairment (which does not meet the criteria for dementia) [138]. Furthermore, 24 years later, a study by Gouhie showed that, at this stage, the evidence available from the published literature does not support the use of piracetam in the treatment of patients with dementia or cognitive impairment in accordance with any of the more specific measures [139]. Although effects have been observed regarding the global impression of change, they confirmed that, despite piracetam’s popularity, there remains a lack of conclusive evidence regarding its effects on cognition and memory.

Studies on Ginkgo Biloba have shown that this drug improves cognitive function, neuropsychiatric symptoms, and consequent reduction of caregiver stress and maintenance of autonomy in patients with age-related cognitive decline, MCI, and mild to moderate dementia [140].

A 30-year history of use of cerebrolysin has demonstrated that this multi-target peptidergic drug with a neurotrophic mode of action exerts its therapeutic effects in Alzheimer’s disease in a long-lasting manner, which may reflect its utility in disease modification [141]. Clinical trials have shown that cerebrolysin is safe and effective in the treatment of Alzheimer’s disease and may enhance and prolong the efficacy of cholinergic drugs, particularly in patients with moderate to advanced Alzheimer’s disease [142].

Importantly, cerebrovascular amyloidosis could also be positively treated with cerebrolysin in a transgenic model of Alzheimer’s disease [143]. In this regard, we question whether xanthurenic acid synthesis modulation could be involved in this therapeutic task.

The beneficial effects of dietary restriction on learning have also been reported [144].

### 7.2. Stochastic Resonance Therapy

Experimental animal studies have shown that exercise has a significant effect on motor activity and learning [145] and exerts regenerative activity [146]. The first therapeutic indication for exercise was introduced as early as 1880 by Jean-Martin Charcot, who described that the use of a rocking chair by Parkinson’s patients led to an improvement in symptoms, particularly tremor and stability [147]. A marked deficit in dopaminergic neurotransmission and its role in the symptoms of PD was reported by Hornykiewicz [101,105]. The therapeutic potential of vibration in PD patients has been confirmed [148,149,150]. Dopamine is significantly affected by exercise [151], and importantly, endogenous KYNA has been shown to control extracellular dopamine levels in rat striatum in vivo [152].

A phenomenon called stochastic resonance, in which oscillations enhance the response of a non-linear system to a weak signal, can affect molecular biological machines and physiological responses. The first report on the mechanism of stochastic resonance was made by Benzi et al. [153], and its significance was applied to a theoretical explanation of the periodic recurrence of the Earth’s ice ages. Interestingly, in biology, stochastic resonance has been experimentally demonstrated in several sensory neural systems, including crayfish, sharks, and crickets [154] as well as in humans. Collins et al. have shown that the tactile sensation of the human fingertip can be enhanced by mechanical vibration [155], and their findings suggest that mechanoreceptors can be affected by stochastic resonance [156].

A major problem for sick people is that a lack of exercise severely limits their quality of life, often accompanied by depression. These people often lack the motivation to stick to an exercise program. Stochastic resonance therapy (SRT) has been established primarily for exercise in healthy individuals but has also shown promise in patients with Parkinson’s disease [157]. SRT allows for non-linear vibration of the body, which has a significant effect on the patient’s motor skills and psyche, without requiring the patient to do anything while standing protected on a vibration plate. Other vibration plates are considered less suitable for therapy. The recommendation to use the SRT came from the patients due to the positive effects on motor function and well-being observed after therapy (Kepplinger observation).

SRT affects tryptophan metabolism by significantly reducing serum L-tryptophan, L-kynurenine, and KYNA levels in healthy human subjects [158]. Importantly, subjects reported increased stability and ease of walking in the hours following SRT. The effect on tryptophan metabolites was time-dependent, and a reduction was measured at 60 min after SRT, indicating a longer-lasting effect. The changes in L-tryptophan metabolism suggest increased incorporation of the amino acid into ongoing biochemical processes. As L-tryptophan significantly crosses the blood–brain barrier, changes in serum may also affect NAD and/or serotonin synthesis not only in the periphery but also in the CNS [5,6,13]. SRT is now being used to rehabilitate patients with several neuropsychiatric disorders, including Parkinson’s disease [157], multiple sclerosis [159], Alzheimer’s disease [160,161], stroke [162], depression, and schizophrenia [163].

Exercise generally has significant effects on learning and long-term potentiation [145], axonal regeneration of sensory neurons [146], restoration of synaptic plasticity [164], and tryptophan metabolism.

### 7.3. Exercise

At present, more and more people are convinced that physical activity plays an important role in well-being and health. The use of different types of exercise (i.e., endurance or resistance exercise) has been extensively studied in relation to tryptophan metabolism. Most of these studies have reported an increase in KYNA metabolism [165,166,167]. These studies were conducted in healthy subjects, and the results obtained are interesting; however, this type of exercise is not always applicable to ill subjects with significantly reduced motor function and likely already increased KYNA formation. There is evidence that regular physical activity also reduces blood–brain barrier permeability through increasing antioxidant capacity, reducing oxidative stress, and having anti-inflammatory effects [168]. Studies have provided evidence that exercise—particularly aerobic exercise—improves cognitive function in several patient populations, including those with Alzheimer’s disease, older people at risk of cognitive impairment, and healthy older adults. The authors suggested that physical activity may play a role in modifying the disease process and preserving cognitive function in Alzheimer’s disease.

### 7.4. Anti-Dementia Drugs Blocking Kynurenic Acid Synthesis

#### 7.4.1. Cerebrolysin

The therapeutic effects of cerebrolysin in dementia and brain injury have been proposed to be due to the neurotrophic and neuroprotective activities of this compound [169,170]. In our previous research, we found that piglet brains presented very low levels of KATs and possessed the ability to block rat KAT activities when present in the reaction mixture for the KAT assay [171]. Notably, piglet brain was used to synthesize a multi-target peptide called cerebrolysin [169]. Furthermore, our study on the effects of cerebrolysin on KYNA formation revealed a significant blocking effect on brain KAT I, II, and III activities [172]. This finding was the basis for the proposal that inhibition of KAT activity could lower KYNA levels at the receptor site, leading to increases in acetylcholinergic and glutamatergic neurotransmission. With this assumption, the increased kynurenine aminotransferase activities observed in the brain of Alzheimer’s patients, HIV-1-infected patients, or in patients with pneumonia enhance not only KYNA [90,92] but probably also xanthurenic acid, which could impair mitochondrial function [75] and might play a role in the induction of a new pathological network and formation of “plaques” [173]. It is believable that the lowering of KAT activities through the use of inhibitors such as cerebrolysin might represent an excellent pharmacological and therapeutic approach for both processes; thus, lowering KAT activity-producing xanthurenic acid might reduce plaque formation and/or lowering of KAT activity-producing KYNA might improve brain activities. Accumulated data indicate that cerebrolysin, as an anti-dementia drug, not only blocks KAT activities in rat and human brains (as shown in an in vitro study) [172] but is also effective in preventing cognitive impairment in different experimental animal models [174,175] and, importantly, significantly ameliorates cerebrovascular amyloidosis in a transgenic model of Alzheimer’s disease [142].

Therefore, in our research, we focus on the phenomena related to “lowering of KYNA and xanthurenic acid levels and ameliorating dementia”.

#### 7.4.2. D-cycloserine

D-cycloserine is a partial agonist at the NMDA receptor site [176], which has a developmental effect on learning [177] and presents anti-amyloidogenic and anti-convulsive properties [178,179]. The most surprising finding is that D-cycloserine induces the inhibition of KYNA formation through blocking KAT I, KAT II, and KAT III in the frontal cortex [180]. This biochemical event may be of potential benefit for CNS treatment of cognition and/or memory in various neuropsychiatric disorders and infectious diseases. At present, D-cycloserine is used clinically for the treatment of drug-resistant tuberculosis [181]. Interestingly, the efficacy of D-cycloserine treatment in schizophrenia has been noted in the presence of neuroleptics in an experimental study [177]. With this in mind, it is compelling that the reduction of dopamine neurotransmission via neuroleptics along with the reduction of KYNA levels via D-cycloserine treatment may represent a promising therapeutic paradigm in patients with schizophrenia. Notably, the most important factor is the dose of D-cycloserine, with treatment with low doses yielding a better therapeutic outcome [182].

Interestingly, Ouednow (2010) made the important point that the search for substances that improve memory reaches its limits in healthy people—where evolution has exhausted the most optimal, healthy brain—but not in sick people [183]. The author also suggested that scientists should therefore focus their efforts on developing substances that improve cognitive performance in people with mental and neurological disorders, rather than trying to improve something that will be difficult to improve: the healthy brain. Cognitive enhancement is therefore possible and desirable but only in appropriately impaired disease groups. Nevertheless, it is reasonable to search for natural drugs that might delay the aging process, as there is significant evidence that the consumption of selected food supports health and good mood [144].

## 8. Herbal Medicines, Prophylaxis, and Protection

Old European reference books on medicinal plants have documented a number of plants, such as Salvia officinalis (sage) and Melissa officinalis (balm), with memory-enhancing properties. Interestingly, Salvia significantly reduced KYNA formation (Baran observation). Thus, we speculate a link between this plant’s ability to inhibit KYNA synthesis and its memory-enhancing properties. Hawthorn berries (“haws”) are used to make wine and jelly and to flavor brandy, and this plant has been used as a remedy for heart problems as well as in the treatment of Alzheimer’s disease; notably, it has been found to block KYNA synthesis [184].

The effects of the Jerusalem Balsam herbal mixture, in terms of blocking the biosynthetic machinery of KYNA synthesis, were also significant. In this study, we demonstrated the high ability of the herbal mixture to influence KYNA formation. We found that Jerusalem Balsam was a dose-dependent and significant inhibitor of KAT II activity in an in vitro assay [185].

We compared the effect of Jerusalem Balsam with that of cerebrolysin and D-cycloserine and found that the inhibitory effect of Jerusalem Balsam on KAT activity was significant and was observed for up to 5 h under the experimental conditions (Figure 7, adapted from [185]). The best inhibitory effect was achieved with Jerusalem balsam, cerebrolysin, and then D-cycloserine. D-cycloserine is used to treat patients showing tuberculosis drug resistance [181]. In this context, we suggest that blockade of KYNA synthesis could be part of a therapeutic measure to increase the efficacy of drugs for viral and/or bacterial diseases. Jerusalem Balsam is also recommended for lung treatment [185]. Hawthorn berries have been recommended for preventive treatment of cardiovascular disturbances [186]. A recently published study demonstrated the improvement of clinical symptoms in COVID-19 patients using a natural herbal medicine, and the authors suggested that it may be effective in treating COVID-19 [133]. Furthermore, evaluation of the adjuvant efficacy of a natural herbal medicine in COVID-19 patients indicated a positive effect on therapeutic success [134].

At present, many plants are used in the treatment of dementia. An interesting review, published by Perry and Howes, described the significant application of medicinal plants in dementia therapy. In addition, the authors speculated that herbs may be used for amelioration of brain aging [187]. Two of the four main drugs currently approved for the treatment of cognitive symptoms in dementia are derived from plant sources (i.e., galantamine and rivastigmine). Thus, natural products are a potential source of other individual compounds that could be developed as drugs for dementia [188]. The potential of ethnomedicinal plants in the treatment of Alzheimer’s disease (AD) is also significant, considering their ability to influence a variety of mechanistic pathways involved in the disease [189]. These plants offer a range of bioactive compounds that can act on key pathological targets, such as amyloid beta aggregation, tau protein hyperphosphorylation, oxidative stress, metal dysregulation, inflammation, and cholinergic deficit.

In the case of amyloid beta aggregation, the reduction in xanthurenic acid formation by KAT inhibitors may be of interest.

The presence of KYNA in plants and foods has also been reported, and various therapeutic approaches have been suggested. KYNA has been found in the leaves, flowers, and roots of medicinal plants such as dandelion (*Taraxacum officinale*), stinging nettle (*Urtica dioica*), and greater celandine (*Chelidonium majus*) [190]. The authors suggested that these plants have therapeutic potential, particularly with respect to the digestive system, and should be considered as new valuable dietary supplements. Studies by Turska et al. [191] have described that KYNA can be synthesized by the gut microbiome and that it can be provided by foods characterized by low levels of KYNA, with the exception of chestnut honey, which is very rich in KYNA. The authors analyzed the health benefits of KYNA-enriched foods. A study by Zgrajka et al. described that the use of herbal preparations containing high levels of KYNA could be considered as an adjunctive measure in the treatment of rheumatoid arthritis and in the prevention of rheumatic diseases [192].

## 9. Snail Memory Model *Helix pomatia*

To find anti-dementia drugs that act as inhibitors of KYNA synthesis, it is necessary to develop a model of memory in which pharmacological agents can be studied in vivo. We believe that the simplicity of the snail as a living organism provides a great advantage due to the lower complexity of biochemical interactions and data interpretation, as has been reported in previous studies [193,194,195]. The snail learns to anticipate the presentation of the food using olfactory cues (i.e., odors), and the tentacle-lowering response is used as an index of learning [196,197]. The snail’s blood–brain barrier is not well developed, allowing drugs to reach the ganglia [198,199]. As in vitro KYNA metabolism in snails has shown that *Helix pomatia* snail tissue is capable of synthesizing KYNA in the CNS as well as in the periphery [20], we considered it reasonable to use *Helix pomatia* to explore the relationship between learning impairment and kynurenine metabolites in a pharmacological study. The pharmacological study showed that L-kynurenine treatment increased brain KYNA levels in vivo, and impaired learning capacity was observed in *Helix pomatia*. At present, anti-dementia drugs and various herbal extracts are under investigation (MS in preparation HB).

## 10. Future Perspectives

In light of the accumulated data, we suggest that modulation of KYNA-induced inhibition of glutamatergic and cholinergic activities may be causally related to the efficacy of both herbs and drugs in the aging process as well as in pathological conditions such as infectious diseases, Alzheimer’s disease, dementia, Down syndrome, and schizophrenia. To protect humans from the adverse effects of aging, prophylactic measures must be taken from an early age. A study by Vohra et al. showed that depletion of KYNA by dietary restriction can lead to activation of the NMDAR of a specific pair of interneurons, playing a critical role in learning [144]. Long-term dietary administration of amino acids may also affect KYNA production by regulating brain L-kynurenine uptake or KYNA synthesis [200]. In the future, using the *Helix pomatia* snail model of memory, we aim to select more herbs and/or drugs that support attention, alertness, and learning and find new meaning for the depletion of the metabolism of KYNA and xanthurenic acid in relation to the glia-depressing factor (GDF).

## Figures and Tables

**Figure 1 biomolecules-15-00074-f001:**
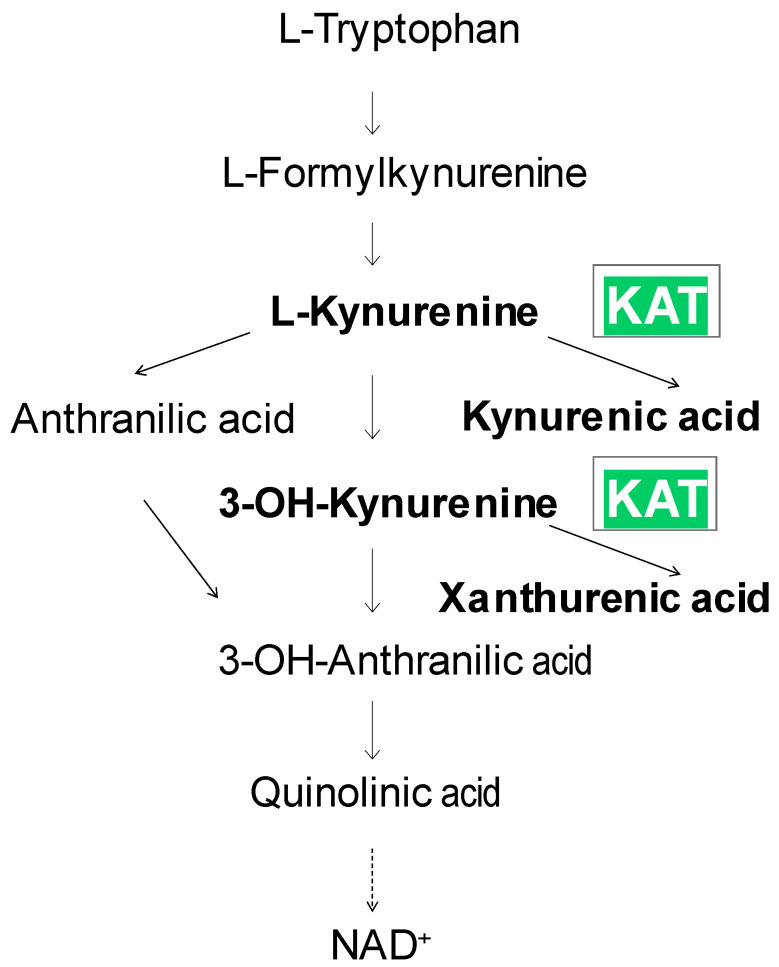
Tryptophan catabolism along the kynurenine pathway.

**Figure 2 biomolecules-15-00074-f002:**
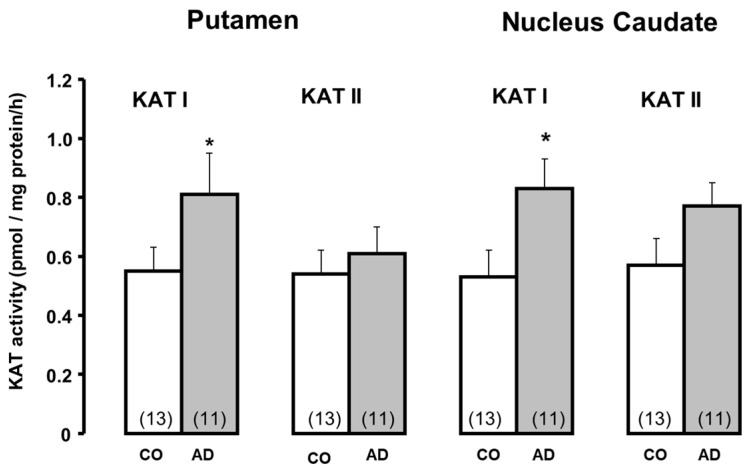
Kynurenine aminotransferase I and II activities in the brain of patients with Alzheimer disease (AD) and controls (CO). Data represent mean ± SEM. Significances: * *p* < 0.05 vs. CO. Data adapted from Baran et al. [90].

**Figure 3 biomolecules-15-00074-f003:**
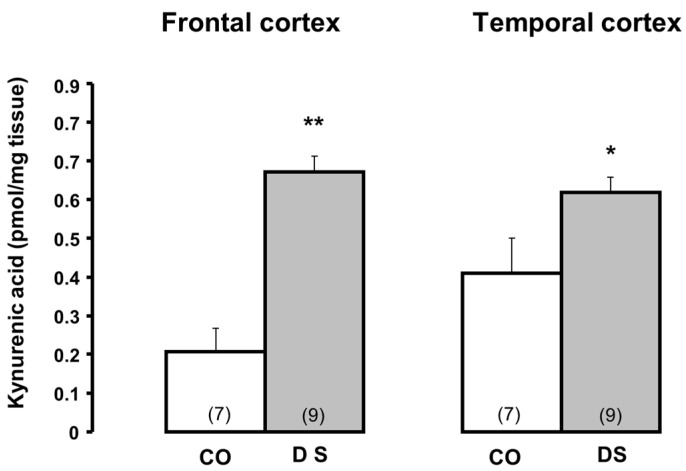
Kynurenic acid levels in the brain of Down syndrome (DS) patients and controls (CO). Data represent mean ± SEM. Significances: * *p* < 0.05; ** *p* < 0.01 vs. CO. Data adapted from Baran et al. [89].

**Figure 4 biomolecules-15-00074-f004:**
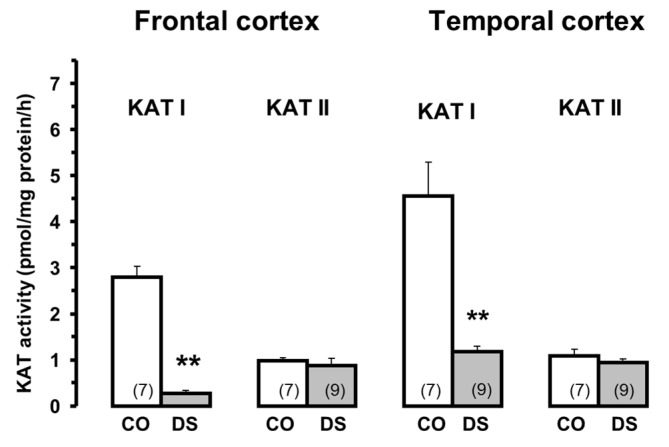
Kynurenine aminotransferase I and II activities in the brain of Down syndrome (DS) patients and controls (CO). Data represent mean ± SEM. Significances: ** *p* < 0.01 vs. CO. Data adapted from Baran et al. [89].

**Figure 5 biomolecules-15-00074-f005:**
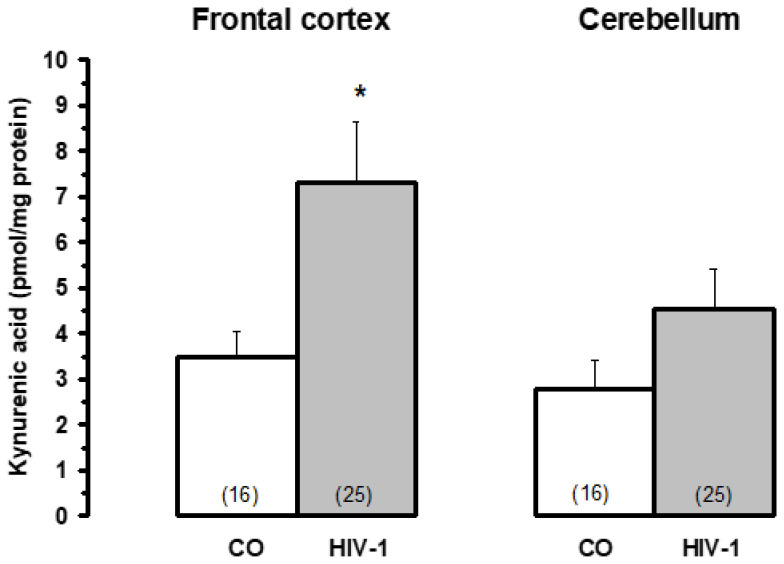
Brain kynurenic acid (KYNA) levels in patients infected with the HIV-1 virus (HIV-1) and controls (CO). Data are mean ± SEM. Significances: * *p* < 0.05 vs. CO. Data adapted from Baran et al. [92].

**Figure 6 biomolecules-15-00074-f006:**
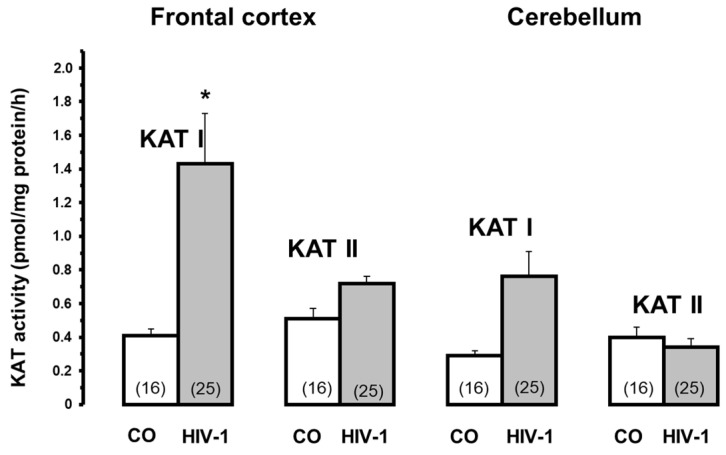
Kynurenine aminotransferase I and II activities in HIV-1-infected brains and controls (CO). Data are mean ± SEM. Significances: * *p* < 0.05 vs. CO. Data adapted from Baran et al. [92].

**Figure 7 biomolecules-15-00074-f007:**
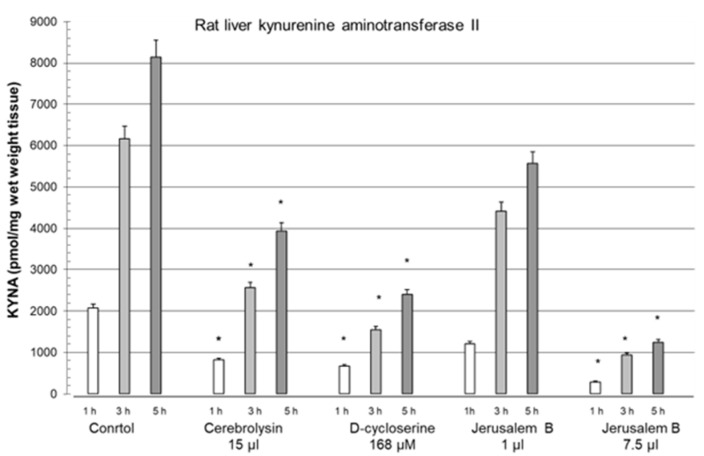
Inhibition of kynurenine aminotransferase II activity in rat liver in the presence of cerebrolysin 15 µL, D-cycloserine 168 µM, and Jerusalem Balsam 1 and 7.5 µL for different times of incubation: 1, 3, and 5 h. Abbreviation: Jerusalem Balsam (Jerusalem B). Data are mean ± SEM. Significances: * *p* < 0.05 vs. CO. Data adapted from Baran et al. [185].

## Data Availability

Not applicable.

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
