# Peer review of "Importance of Modulating Kynurenic Acid Metabolism—Approaches for the Treatment of Dementia"

_biomolecules, 2025, doi:10.3390/biom15010074_

Round 1
Reviewer 1 Report
Comments and Suggestions for Authors
- This is a comprehensive review done by a group of experts in the field of Neurochemistry.
- Figures 2, 3, 4, 5, and 6 in the article are adapted from other papers. The original paper is mentioned only in the text, but not below the corresponding figure.
- In the Abstract, the experimental model of Helix Pomatia snail is mentioned, but in the text in mentioned very shortly (rows 441-448). It could be an interesting experimental model to mention with more details.
- Exercise and physical activity are mentioned in the Abstract as methods with impact on the kynurenic acid metabolism, but in the text only Stochastic Resonance Therapy is mentioned. It is possible that other physical exercises could influence the aging process and kynurenine methabolism., and it may be interesting to mention that also.
- Please provide the detailed name of EMCV (page 3, row 121)
Author Response
Thank you for the suggestions. Please see the attachment for specific responses.

Reviewer 2 Report
Comments and Suggestions for Authors
The review manuscript “Importance of modulating kynurenine metabolism – approaches for the treatment of dementia” aims to review experimental studies including new treatments and compounds that can block kynurenine aminotransferase as a treatment for memory impairment and dementia. Although the subject is interesting, the approach of the present review is ambiguous and lacks of sufficient information.
Considering the literature review nature of this manuscript, several parts of the text should include more detailed information.
For example, in line 55-56 the authors mention “In rat and human peripheral tissues, four types of proteins are capable of catalyzing the kynurenine-2-oxoacid transamination reaction to form KYNA” however, they do not explain exactly which proteins are being referred to.
Some typos such as “catalysing” instead of “catalyzing” are present in the same lines.
Some ideas are written in a confusing way i.e. “ KAT I, characterised by a high pH optimum of 9.6” , however, the pH optimum is not an intrinsic characteristic of this enzyme but of its catalytic activity. Another example “KATs appear to have the ability to change their chemical properties and presumably their actions under physiological and pathological conditions”, What chemical properties do they refer to? An how they change under physiological vs pathological conditions?
Line 92, please explain in detail how KYNA increases the expression of α-4-ß-2nAChRs.
There is extensive evidence about the neuroprotective properties of KYNA as well as its function on NMDA and α-7nACh receptors, so more information could be included in this section.
KYNA and “significant abnormalities”, it is unclear what this sentence refers to.
Line 110 and 111, it should be clarified at what stage of development, under physiological conditions?
Lines 120-124, are confusing. The statement ”It would be therefore be reasonable…” has no congruence with previous sentences.
Line 138 What does spontaneous synthesis of KYNA stands for?
Line 168-169 How is that the dramatic decrease in KYNA on the day of birth suggests that KYNA is involved in biochemical processes/events that regulate lung function?
Section Glutamatergic and acetylcholinergic activity and dementia seem incomplete.
Line 326, please explain what glial depressing factor refers to.
Overall, the manuscript of this literature review lacks the necessary information and clarity in several sections that are addressed.
The lack of continuity of ideas and congruence from one sentence to the next makes this review difficult to read. Additionally, the informal style in which some ideas are expressed, such as “I once asked Prof Kido what was more important to know about a molecule, the effect of the molecule or the mechanism of action of the molecule. Prof Kido's answer was - the effect” should be modified.
Comments on the Quality of English LanguageMinor editing of English language required.
Author Response

(The authors gave the same response as above.)

Reviewer 3 Report
Comments and Suggestions for Authors
This review summarized the metabolism and pharmacological roles of kynurenine acid on dementia and related neurological conditions.
Follows are the comments:
1.Line 108: 3. Kynurenic acid and significant abnormalities: This title is suggested to be more detailed.
2.Line 237-238: Suggest to write formally. “The lack of serotonin lowering in the kainic acid model was one of the 237 miracles of my study and work," commented Prof Oleh Hornykiewicz, my PhD supervisor at the time.”
3.Figures 3-7 were adapted from other published papers. Any copyright issue?
4.Line 481-482: Suggest to write formally. “I once asked Prof Kido what was more important to know about a molecule, the effect of the molecule or the mechanism of action of the molecule. Prof Kido's answer was - the effect.”
5. the Helix pomatia snail model of memory is suggested to be described briefly.
6.Part5. The role of Kynurenic acid regarding “Glutamatergic and acetylcholinergic activity and dementia” should be added or re-organized into this part.
Comments on the Quality of English LanguageSome parts are suggested to write formally.
Author Response

(The authors gave the same response as above.)

Reviewer 4 Report
Comments and Suggestions for Authors
The review predominantly focuses on kynurenic acid, rather than providing a comprehensive overview of the kynurenine pathway. It is indeed surprising that IDO is not even mentioned once and other metabolites such as quinolinic acid -which are highly involved in neurodegeneration are only briefly mentioned. As such, the title of the manuscript is misleading.
Moreover, the link between KYNA and alpha 7 nicotinic receptors are often mentioned but the binding of beta-amyloid to this receptor, which is believed to be an important mechanism in Alzheimer's disease has been missed out.
Some of the recent work on the kynurenine pathway and neurodegenerative disorders can be mentioned (eg. PMID's: 31055163, 23630570, 30980800, 31607909, 38706427).
Lastly, it is difficult to summarise the manuscript or get a "main message". It seems more of an overview and the authors' own research history of kynurenic acid, without any novel speculations or research ideas supported by the existing literature.
Comments on the Quality of English LanguageThe writing is mostly acceptable without major flaws and errors. However, it would not be counted as high-quality language usage.
Author Response

(The authors gave the same response as above.)

Round 2
Reviewer 1 Report
Comments and Suggestions for Authors
All suggestions have been accomplished.
Comments on the Quality of English Language
Minor English editing needed
Author Response
Thanks for your comments. All suggestions were implemented. English version was requested and English version was made.Reviewer 2 Report
Comments and Suggestions for Authors
I carefully reviewed the authors' response to my comments as well as the corrected version of the manuscript and the comments I made during the first review were considered, resolved, and added to the new version of the manuscript and I recommend this version for publication.
Author Response
Thanks for your comments. All suggestions were implemented. English edition was requested and English edition was made.
Reviewer 4 Report
Comments and Suggestions for Authors
The change in title is welcomed and fits the manuscript better. Nevertheless, beyond the title, I still fail to see the main message or aim of this manuscript. Dementia seems to make-up only a relatively small part of the manuscript and a lot of unrelated information is provided. In many parts (ie. the discussion of pH and piglet livers) very long bits of information are given, without relating it to an argument or main message. The snail model of memory is interesting, but apart from the the observation of memory-related behaviours and KYNA, it is not clear what makes this a model of dementia, memory, or kynurenine pathway deficits. It is (almost) completely detached from the rest of the manuscript and comes out of nowhere at the very end. Unless this manuscript gets majorly concentrated with a specific aim or argument, it remains a literature review for KYNA by a group of experts. While the usefulness of such a review is debatable, I respect the authors' choice of writing it. If this is the case, it should not be treated as a review article for KYNA and dementia.
Comments on the Quality of English LanguageAt parts the manuscript reads as daily English rather than scientific writing.
Author Response
Thanks for your comments. I try to do the best with my first revision, but the 3rd Reviewer was not sutysfy. Probably other people will do it differently and I have no problem with that. I am convinced that the paper has improved a lot and is as I would like to see it. I hope the reviewer will accept it.